# Sex Differences in the Neuroendocrine Stress Response: A View from a CRH-Reporting Mouse Line

**DOI:** 10.3390/ijms252212004

**Published:** 2024-11-08

**Authors:** Krisztina Horváth, Pál Vági, Balázs Juhász, Dániel Kuti, Szilamér Ferenczi, Krisztina J. Kovács

**Affiliations:** 1Laboratory of Molecular Neuroendocrinology, Institute of Experimental Medicine, Hungarian Research Network, 1083 Budapest, Hungary; horvath.krisztina@koki.hun-ren.hu (K.H.); juhasz.balazs@koki.hun-ren.hu (B.J.); kuti.daniel@koki.hun-ren.hu (D.K.); ferenczi@koki.hu (S.F.); 2Nikon Center of Excellence, Institute of Experimental Medicine, Hungarian Research Network, 1083 Budapest, Hungary; vagi.pal@koki.hun-ren.hu; 3János Szentágothai Doctoral School of Neurosciences, Semmelweis University, 1085 Budapest, Hungary

**Keywords:** Crh-IRES-Cre, tdTomato, sexual dimorphism, psychological stress, c-Fos, physiological stress, predator odor

## Abstract

Corticotropin-releasing hormone (CRH) neurons within the paraventricular hypothalamic nucleus (PVH) play a crucial role in initiating the neuroendocrine response to stress and are also pivotal in coordination of autonomic, metabolic, and behavioral stress reactions. Although the role of parvocellular CRH^PVH^ neurons in activation of the hypothalamic–pituitary–adrenal (HPA) axis is well established, the distribution and function of CRH-expressing neurons across the whole central nervous system are less understood. Stress responses activate complex neural networks, which differ depending on the type of stressor and on the sex of the individual. Because of the technical difficulties of localizing CRH neurons throughout the rodent brain, several CRH reporter mouse lines have recently been developed. In this study, we used Crh-IRES-Cre;Ai9 reporter mice to examine whether CRH neurons are recruited in a stressor- or sex-specific manner, both within and outside the hypothalamus. In contrast to the clear sexual dimorphism of CRH-mRNA-expressing neurons, quantification of CRH-reporting, tdTomato-positive neurons in different stress-related brain areas revealed only subtle differences between male and female subjects. These results strongly imply that sex differences in CRH mRNA expression occur later in development under the influence of sex steroids and reflects the limitations of using genetic reporter constructs to reveal the current physiological/transcriptional status of a specific neuron population. Next, we compared the recruitment of stress-related, tdTomato-expressing (putative CRH) neurons in male and female Crh-IRES-Cre;Ai9 reporter mice that had been exposed to predator odor. In male mice, fox odor triggered more c-Fos in the CRH neurons of the paraventricular hypothalamic nucleus, central amygdala, and anterolateral bed nucleus of the stria terminalis compared to females. These results indicate that male mice are more sensitive to predator exposure due to a combination of hormonal, environmental, and behavioral factors.

## 1. Introduction

Corticotropin-releasing hormone (CRH) plays a pivotal role in the stress response system [1]. Parvocellular CRH neurons in the hypothalamic paraventricular nucleus (PVH) initiate and modulate the neuroendocrine hypothalamic–pituitary–adrenal (HPA) stress axis by releasing CRH into the hypophyseal portal circulation [2,3]. However, CRH’s influence extends beyond the neuroendocrine stress response; it is also critical in mediating autonomic, metabolic, and behavioral reactions to stress [3,4,5].

In addition to the PVH, CRH-expressing neurons are widely distributed throughout the central nervous system (CNS) [6,7,8,9]. While the structure and function of neurosecretory PVH CRH neurons are well characterized [10,11,12], much less is known about the roles of CRH cells located outside the hypothalamus.

Recent advancements in genetic engineering, particularly through the application of Cre-lox technology, have enabled a detailed exploration of CRH-expressing neurons across the central nervous system. Among the mouse models utilizing this technology, the Crh-IRES-Cre;Ai14 tdTomato mouse, as reported by Chen et al., exhibited an almost complete overlap between the native peptide and the tdTomato reporter. However, their investigation was confined to the paraventricular nucleus of the hypothalamus, the bed nucleus of the stria terminalis, the amygdala, and the hippocampus [13]. Although several earlier studies have employed the Ai14 reporter line [13,14,15], we have found that the Ai9 reporter is more suitable for colocalization studies due to its moderate expression levels of the fluorescent marker gene [9].

Psychological stressors, such as restraint and exposure to predator odor, along with physiological stressors, such as ether exposure, hypertonic saline, and lipopolysaccharide administration, activate distinct brain regions [9,16]. Systemic (physiological) stressors activate afferents originating from the brainstem, specifically the solitary tract, locus ceruleus, and ventrolateral medulla, transmitting information directly via catecholaminergic (noradrenergic) pathways to the PVH. In contrast, psychological stressors operate through a range of indirect inputs. For example, limbic structures—including the lateral septum, hippocampus, and central and medial amygdala—modulate the stress axis by interacting with other brain regions, primarily influencing neurons in the amygdala, the bed nucleus of the stria terminalis, or the peri-PVH area [11,17]. According to the literature, different stressors provoke distinct patterns of neuronal activation, i.e., within the PVH itself [16,18] and within the BNST and amygdaloid complex.

Sex-based differences in stress responses have been the subject of extensive research, demonstrating that males and females exhibit distinct neurobiological profiles when faced with stressors. These differences are influenced by variations in hormonal levels, patterns of neural activation, and differences in behavioral coping strategies [19,20,21,22,23,24,25,26], but the field needs more research.

The diversity of neural networks engaged during different types of stress reactions, coupled with the observed sex-related differences in stress responses, has prompted us to investigate whether stress-related neurons and CRH cells are recruited in a stressor- or sex-specific manner, both within and outside the hypothalamus. In this study, we identified neural circuits that specifically respond to psychological or physiological stressors and assessed potential sex-based differences in CRH cell distribution by quantifying the neurons in various stress-related brain regions.

## 2. Results

To investigate the distribution of CRH-expressing neurons in the adult mouse brain, Crh-IRES-Cre;Ai9 reporter mice were used.

CRH Cre-driver mouse lines [27] serve as an effective detection strategy. In Crh-IRES-Cre (B6(Cg)-Crh^tm1(cre)Zjh^/J) mice, the 3′ untranslated region of the corticotropin-releasing hormone (*Crh*) locus harbors an internal ribosome entry site and a Cre recombinase. This arrangement directs Cre expression via the endogenous CRH promoter/enhancer elements (Figure 1).

The combination of Cre-driver mouse lines with various reporter constructs provides a straightforward approach to visualize CRH-expressing neurons throughout the central nervous system [6,14,27]. When Crh-IRES-Cre animals are bred with mice that possess loxP-flanked sequences, Cre-mediated recombination occurs in the CRH cells of the offspring. The Ai9 strain (B6;Cg-Gt(ROSA)26Sor^tm9(CAG-tdTomato)Hze^/J) serves as a Cre reporter strain. Therefore, crossing homozygous Crh-IRES-Cre and Ai9 mice results in selective fluorophore expression within the CRH neurons in the F1 heterozygous Crh-IRES-Cre;Ai9 offspring (Figure 2). Using this genetic approach, all CRH-expressing neurons are labeled in red. It is also conceivable that all neurons are labeled that have ever expressed the *Crh* gene throughout development, independent of present *Crh* gene activity.

### 2.1. Validation of Crh-IRES-Cre;Ai9 Mice

In earlier studies [9], we established that Cre-negative control mice exhibit no tdTomato signal throughout the brain. Additionally, we observed greater than 80% overlap between tdTomato expression and CRH mRNA-positive neurons (detected using RNAscope) in the hypothalamic paraventricular nucleus (PVH) of the mice. Since tdTomato expression is driven by the constitutively active CAG promoter, it is not influenced by the current levels of CRH expression. This was confirmed by our observation that neither adrenalectomy nor acute stress exposure altered tdTomato signal intensity. All these findings confirm that Crh-IRES-Cre;Ai9 mice are suitable for the subsequent experiments.

### 2.2. Comparison of tdTomato+ (CRH) Neurons in Female and Male Crh-IRES-Cre;Ai9 Mice

To reveal potential sex differences in the stress response system, we quantified tdTomato-positive cells in 23 brain regions in female and male Crh-IRES-Cre;Ai9 mice. Based on our previous experiment [9], we selected brain regions that play a critical role in HPA axis regulation or are important members of the limbic system with a significant role in responding to psychological or physiological stressors.

We found that the number of tdTomato+ neurons within the bed nucleus of the stria terminalis, anterior division, dorsal part (BSTad), was slightly higher in male mice than in female animals. However, no sex-dependent differences were observed in the number of CRH neurons in any of the other examined regions, namely, the piriform area (PIR); anterior cingulate area (ACA); Ammon’s horn (CA); dentate gyrus (DG); field CA3 (CA3); lateral septal nucleus (LS); central amygdalar nucleus (CEA); medial amygdalar nucleus (MEA); bed nucleus of the stria terminalis, anterior division, ventral part (BSTav); bed nucleus of the stria terminalis, anterior division, oval nucleus (BSTov); bed nucleus of the stria terminalis, anterior division, anterolateral area (BSTal); bed nucleus of the stria terminalis, posterior division (BSTp); bed nucleus of the anterior commissure (BAC); central medial nucleus of the thalamus (CM); paraventricular nucleus of the thalamus, anterior (PVTa); periventricular region (PVR); medial preoptic nucleus (MPN); anterior hypothalamic nucleus, posterior part (AHNp); ventromedial hypothalamic nucleus (VMH); supramammillary nucleus (SUM); periaqueductal gray (PAG); and paraventricular nucleus of the hypothalamus (PVH). Diagrams in Figure 3 show the quantified tdTomato+ cell counts and representative microscopic images from the BSTad, ACA, CEA, and PVH.

### 2.3. Psychological and Physiological Stressors Induce Differential Neuronal and tdTomato+ (CRH) Activation in the Mouse Brain

In our previous research [9], we applied restraint and predator odor exposure as psychological stressors, and we applied ether exposure, intraperitoneal lipopolysaccharide administration, and hypertonic saline injection as physiological stressors. Summarizing the activation patterns published in that study, we found that the elevations in neuronal and tdTomato+ cell activation induced by psychological and physiological stressors (compared to control animals) were significantly different across various distinct brain regions. Figure 4 highlights the areas where either psychological or physiological stressors resulted in greater increases in activation levels, along with their corresponding significance. However, not all brain regions are presented, but rather the areas that are directly or closely related to the limbic system.

Psychological stressors may precipitate various neuropsychiatric disorders, including anxiety, depression, and post-traumatic stress disorder (PTSD). Because these stress-related abnormalities are more prevalent in females than males, we next assessed the sex difference between recruitment of CRH neurons in response to psychological stress. Predator odor exposure has been selected as an animal model because it is a severe psychological/traumatic stressor that initiates innate defensive behavior, results in lasting hyperarousal.

### 2.4. Sex Differences in the Predator-Odor-Induced Neuronal Activation of tdTomato+ CRH Neurons in the Mouse Brain

To investigate the sex difference in recruitment of CRH neurons’ response to predator odor, male and female Crh-IRES-Cre;Ai9 mice were exposed to synthetic fox odor (2-methyl-2-thiazoline, 2-MT). Activated neurons were identified by nuclear c-Fos immunostaining, putative CRH neurons displayed tdTomato red fluorescence in the cytoplasm, and activated CRH neurons expressed both markers. Male mice were more reactive to fox odor exposure than females: more c-Fos positive cells were counted in the PVH, CEA, and BSTal areas in males than in females. Furthermore, we found more activated tdTomato+ (putative CRH) cells in the BSTal region of male mice than female mice (Figure 5).

## 3. Discussion

In our earlier study [9], we demonstrated that male Crh-IRES-Cre;Ai9 mice serve as an appropriate model for mapping corticotropin-releasing hormone (CRH) cells and identifying activated CRH neurons through c-Fos and tdTomato colocalization. A significant overlap between tdTomato expression and CRH mRNA-positive neurons was found in stress-responsive hypothalamic and extrahypothalamic areas of the mouse brain. In this work, we also used Crh-IRES-Cre;Ai9 reporter mice for comprehensive analysis of the number, distribution, and stress-specific recruitment of CRH neurons in male and female mice.

To enhance the reliability and comparability of our mapping results, we defined and selected all examined brain areas based on the online Allen Brain Atlas, which provides a globally standardized reference for brain structure and function [28].

In this study, we investigated whether sex-related differences in the prevalence of psychiatric disorders could arise from variations in CRH distribution, specifically within limbic and stress-related regions.

Using Crh-IRES-Cre;Ai9 mice, we have previously [9] identified 95 brain areas with significant tdTomato/stress-induced c-Fos expression. Among these, we selected 23 areas and counted the number of putative CRH neurons based on their red fluorescence in the brains of male and female mice. In contrast to our hypothesis, quantification of tdTomato expressing profiles revealed only one difference between male and female mice. Males had slightly higher number of CRH-positive neurons in the dorsal part of the anterior division of the bed nucleus of the stria terminalis (BSTad) compared to females. However, no significant sex-related differences were observed in other regions such as the PVH, central and medial amygdala, or hippocampus.

The BNST is an exceptionally complex brain region, consisting of 18 subnuclei and 41 transcriptionally distinct neuronal populations in mice [29]. This intricate structure underlies its broad range of functions, including integration of information related to hedonic valence, mood, arousal, motivation, and emotional processing, which collectively shape motivated behaviors, stress, anxiety, and social interactions [30,31]. Each subdivision within the BNST is associated with specific roles [31]. Regions contain a diverse array of neurons that vary in their electrophysiological properties, morphology, spatial organization, neuropeptide content, and synaptic connectivity, all of which contribute to their unique function [30]. The BNST expresses a wide variety of neuropeptides, many of which are stress-sensitive, including CRH, urocortin, pituitary adenylate cyclase activating polypeptides, vasoactive intestinal peptide, neuropeptide Y, neurotensin, somatostatin, oxytocin, and vasopressin [32]. Functionally, the BNST is often categorized into anterior, posterior, dorsal, ventral, medial, and lateral subdivisions. Both the anterior and posterior divisions play a key role in autonomic control [30,33]. Furthermore, the anterior division is critical for mediating responses to social threats and stress-induced impairments in social behavior. Innate fear is also linked to this region [30,34,35]. In contrast, the posterior division plays a pivotal role in regulation of sex-specific social behaviors, such as aggression, mating, and parental care [30,33]. The anterolateral BNST (BSTal) has been extensively studied for its involvement in the regulation of stress and anxiety [30]. The BSTal is also associated with regulation of startle responses [36].

The BNST is a highly sexually dimorphic brain region, with notable differences in structure and function between males and females [37]. Literature indicates that social behavior is regulated by vasopressin cells in the BNST in a sexually dimorphic manner [38]. The posterior BNST has been identified as the most sexually dimorphic sub-region [39]. Uchida et al. (2019) found that adult female mice have significantly more CRH neurons in the oval nucleus of the BNST (BSTov) and in the BSTal than males. Interestingly, this significant sex difference in CRH distribution could not be observed on postnatal day five. To distinguish the influence of sex steroids on CRF neuron numbers, Uchida and colleagues performed gonadectomy on both male and female mice. Ovariectomized females exhibited significantly fewer CRH-expressing neurons than proestrous females in both the BSTov and the BSTal, while orchiectomized males had significantly more CRH cells than intact males in the BSTov [40]. The observation that sexually dimorphic expression of CRH in these subnuclei was regulated by gonadal steroids aligns with the earlier findings of Fukushima et al. (2013), who reported that testosterone exposure during the critical developmental period reduced CRH-immunoreactive neurons in the BNST of female rats [41]. Similar hormonal influences on the sexual dimorphism of the BNST have also been noted in human studies [42].

The discrepancy between our results and those reported in the literature regarding sex differences in CRH cell numbers and mRNA expression in the BSTad can be attributed to the differences in experimental methodology. In our study, CRH neurons were identified using Cre-reporter transgenic mice (Crh-IRES-Cre;Ai9), in which a CAG-LoxStopLox-tdTomato construct is integrated into Gt(Rosa)26Sor locus [43]. Since Cre recombinase and the CAG promoter are active throughout ontogenesis, it is likely that neurons active during embryonic development or in early life stages, but not in adulthood, also express the marker. Thus, although the rising testosterone levels during the critical developmental period may have reduced the number of CRH neurons in the BNST of males in the animal model, this reduction remain undetectable in our study. Our results imply that CRE-based reporter mouse lines (i.e., Crh-IRES-Cre;Ai9) identify all neurons that have ever expressed the gene of interest (CRH) and are useful for anatomical localization studies; however, they are not suitable for assessing the current transcriptional activity of that gene.

Using in situ hybridization histochemistry, studies found higher basal CRH mRNA expression in the PVH of females than in that of males [44,45]. By contrast, Sterrenburg et al. did not find significant differences between males and females [46] in CRH mRNA levels under non-stressful conditions. It should be noted, however, that in Crh-IRES-Cre;Ai9 (and in other Cre-based reporter lines), all neurons in which the *Crh* gene has ever been expressed throughout the entire course of development will express tdTomato irrespective of the current physiological status. Ontogenetic analysis of CRH mRNA in fetal and neonatal mouse brains revealed CRH expression as early as embryonic day 13.5 in the PVH, Barrington’s nucleus, the olivary complex, and the amygdala, while cortical expression occurs only after birth [47]. Together, these findings strongly indicate that sexually dimorphic differences seen in CRH expression develop under the influence of sex steroids and cannot be reliably detected using CRH reporter mouse lines.

Stressors are classified into two categories: physiological stressors, which pose a risk to physical integrity, and psychological stressors, which threaten mental well-being [48]. Physiological stress is often defined as the body’s reaction to physical stressors, such as illness, injury, condition, and intense exercise [49]. These physiological challenges directly impact homeostatic parameters, with their effects mediated through well-characterized receptor systems. The afferent pathways involved include viscerosensory pathways that activate subcortical autonomic circuits and influence stress-related motoneurons with minimal cortical involvement [16]. In contrast, psychological stress refers to the body’s response to strong emotional experiences, such as anxiety, sadness, fear, and anger [50]. Psychological stressors engage a range of somatosensory and nociceptive afferent pathways, with the processed information traveling through intricate cortical and limbic circuits that integrate cognitive, learned, and emotional aspects [16]. Physiological stress typically elicits a motor fight-or-flight response, while psychological stress places greater emphasis on emotional regulation and goal-directed behavior, which is accompanied by reduced reward processing [51]. Although both types of stressors trigger complex adaptive responses primarily through the activation of the core stress circuit—a network of similar, commonly activated brain structures—the degree of neural activation varies across and within different brain regions [52,53,54,55,56]. In addition to these variations in core stress system activation, specific brain regions responsive to distinct stressors have also been identified [26,57,58]. Our findings align with previous studies, highlighting that the neural circuits engaged by psychological stressors (e.g., restraint and predator odor) significantly differ from those activated by physiological stressors (e.g., ether exposure, lipopolysaccharide injection, and hypertonic saline administration). In our earlier research, we observed that physiological stress resulted in comparatively limited neural circuit activation, whereas psychological stressors were associated with more widespread neuronal activation, particularly in the cortical plate (including the olfactory areas, isocortex, and hippocampal formation), thalamus, and hindbrain (pons and medulla) [9].

In this study, we investigated neuronal and corticotropin-releasing hormone cell activation in response to the specified physiological and psychological stressors [9], with a particular focus on the limbic system. Our findings revealed significant differences in the elevation of neuronal and tdTomato+ cell activation induced by psychological versus physiological stress across various brain regions. Specifically, physiological stressors resulted in a markedly greater increase (relative to the control groups as measured by fold change) in neuronal activation levels within the central amygdalar nucleus (CEA) and the posterior division of the bed nucleus of the stria terminalis (BSTp) compared to psychological stressors. In contrast, the regions specifically responsive to psychological stress included the anterior regions of the bed nucleus of the stria terminalis (BNST), the bed nucleus of the anterior commissure (BAC), hippocampal regions, the lateral septal nucleus (LS), the medial preoptic nucleus (MPN), the periaqueductal gray (PAG), and the supramammillary nucleus (SUM).

The central amygdaloid nucleus plays a crucial role in processing sensory and physiological information, and it also influences learning and motivated behaviors related to both reward and threat. [59]. Interestingly, plasma corticosterone levels were significantly lower in rats with temporary CEA blockades exposed to physical stress than in animals with functional CEA that were also physically stressed. In contrast, no significant difference in corticosterone levels was observed between the temporarily CEA-blocked animals exposed to psychological stress and those with functional CEA also being subjected to psychological stress [60]. This highlights the unique role of the CEA in the physiological stress response. Regarding the hippocampal regions, which play a key role in regulating stress-related processes and memory, research has demonstrated that psychological stress can impair spatial working memory [61]. Additionally, reduced hippocampal volume has been identified as a predictor of increased vulnerability to psychological trauma [62].

Here we demonstrated that in addition to more limbic structures contributing to the orchestration of psychological stress reactions, the elevation of neuronal activation in most regions is also significantly higher with psychological stress than with physiological stress. These findings are crucial, as they suggest that psychological stress may uniquely contribute to mental health issues owing to its profound overall impact on the limbic system. Accordingly, significant role has been identified for nearly all limbic regions engaged by psychological stress in psychiatric disorders. For example, the LS has been implicated in anxiety, depression, and eating disorders [63]. The SUM may contribute to sleep disorders, memory impairment, and anxiety [64,65,66]. Dysfunction in the BNST has been associated with anxiety, addiction, and post-traumatic stress disorder (PTSD) [67,68]. Altered activity in the CEA is linked to anxiety and alcohol-related behaviors [69]. Hippocampal regions are associated with depression and Alzheimer’s disease [70]. Finally, the MPN is implicated in mood and sleep disorders [71,72], while the bed nucleus of the anterior commissure (BAC) is involved in mood disorders [73]. Cognitive and limbic dysfunctions are core symptoms of many mental disorders [74].

Certain psychiatric disorders (i.e., anxiety and depression) are more prevalent in women than in men [45], potentially due to sex-based differences in the underlying stress circuitry. Numerous studies have highlighted these differences, showing that males and females display distinct neurobiological profiles in response to stressors. Differences in stress responses are influenced by hormonal variations, neural activation patterns, and behavioral strategies. Cortisol secretion patterns reveal that females often experience a more pronounced and prolonged cortisol response to stress, while males typically show a sharper, shorter spike [19]. This difference can be attributed to sex hormones, with estrogen linked to increased sensitivity to stress [20] and testosterone associated with reduced sensitivity and more aggressive behaviors [21]. Neural activation patterns further illustrate these differences; men exhibit more pronounced stress responses in the prefrontal cortex regions, habenula, and hippocampal subfields, while women display heightened responses in limbic and striatal areas [23,25,26]. Behavioral responses also vary: males typically adopt fight-or-flight strategies, while females often use tend-and-befriend mechanisms to seek social support [23]. These coping strategies contribute to the higher prevalence of anxiety and depression in women after stressful events, whereas men may show increased rates of substance abuse or aggressive behavior [24]. Similar differences in stress-induced neuronal activity patterns were detected in preclinical models. For instance, male mice tend to express less c-Fos (an immediate-early gene product related to neuronal activation) in the hypothalamus in response to restraint (or other psychological challenges); they have dampened activation of HPA axis activity, and corticosterone returns more easily and quickly to the baseline level in males than in females. By contrast, here, we found increased c-Fos expression and increased number of activated CRH neurons in response to predator odor exposure. Indeed, predator odor is one of the psychological stressors that provoke heightened behavioral and hormonal responses in males compared to females. Our recent findings reveal that some of the differential activation pattern of CRH neurons in the male brain is related to differences in hormonal and behavioral responses to predator odor in male mice.

Corticotropin-releasing hormone neurons in the parvocellular region of the paraventricular hypothalamus (PVH) are pivotal in initiating the neuroendocrine stress response, while the broader CRH system in the brain modulates autonomic and behavioral aspects of stress [5]. Central CRH delivery influences functions such as locomotion, arousal, anxiety, reward processing, learning, and memory [75,76,77]. External and internal threats commonly mobilize both the hypophysiotropic and brain CRH systems. In our previous research [9], we found that stressors with strong psychological components activated a more widespread population of brain CRH neurons than systemic challenges. These findings suggest that psychological stress may uniquely contribute to mental health disorders through its broader influence on the limbic system, as previously discussed, with the added understanding that CRH neurons also play a crucial role in mediating this impact. This aligns with findings that link heightened CRH activity to several psychopathologies, such as depression, anxiety [78], PTSD [79], anorexia [80], and panic disorder [81]. Although the role of parvocellular CRH^PVH^ neurons is well established, further research is needed to clarify the exact function of other CRH clusters across the central nervous system.

In summary, our present study reveals that tdTomato marker expression in Crh-Ires-CRE;Ai9 mice does not precisely mirror the sex differences in CRH mRNA, which develop in the brains of adult male and female mice under the influence of sex steroids during ontogenesis and after puberty. The presence of the CAG promoter in Crh-Ires-CRE;Ai9 mice results in constitutive expression of the tdTomato marker gene, irrespective of the transcriptional status of the *Crh* gene. Nevertheless, we found more c-Fos immunoreactive neurons in the brain’s stress circuitry in predator-odor-exposed males than females. Furthermore, more CRH neurons express the activation marker c-Fos in the male BNST than in that of females, indicating sex differences in organization of neuroendocrine, autonomic and behavioral responses to predators.

## 4. Materials and Methods

### 4.1. Animals

Crh-IRES-Cre (B6(Cg)-Crhtm1(cre)Zjh/J; stock number: 012704) and Ai9 (B6;Cg-Gt(ROSA)26Sortm9(CAG-tdTomato)Hze/J; stock number: 007905) mice were originally obtained from The Jackson Laboratory. Both lines were maintained as homozygotes at the Transgenic Facility of Institute of Experimental Medicine. Crh-IRES-Cre;Ai9 reporter mice were generated by crossing these homozygote pairs, and in F1 heterozygotes, Cre-mediated recombination resulted in tdTomato fluorophore expression in CRH neurons. Adult female and male mice, aged 10–12 weeks, were used in this experiments. The animals had ad libitum access to standard laboratory chow and water and were housed under controlled conditions: temperature 21 ± 1 °C, humidity 65%, light intensity 400 lx, and a 12-h light/dark cycle with lights on at 07:00 AM.

All experiments complied with the ARRIVE guidelines and were performed in accordance with the European Communities Council Directive (86/609 EEC), EU Directive (2010/63/EU), and the Hungarian Act of Animal Care and Experimentation (1998; XXVIII, Sect. 243/1998). The procedures were approved by the Animal Care and Use Committee of the Institute of Experimental Medicine, Hungarian Academy of Sciences (permit number: PEI/001/29-4/2013).

### 4.2. Acute Predator Odor Exposure

Mice were exposed to a synthetic analog of the fox anogenital product 2 methyl-2-thiazoline, or 2-MT [34] (CAS 2346-00-1, Santa Cruz Biotechnology, Inc., Dallas, TX, USA), in a covered transparent plexiglass arena (40 × 20 × 20 cm) under a fume hood. One day prior to the experiment animals, were habituated for 20 min to the test box, which contained an empty Eppendorf tube lid placed in the corner. On the day of exposure, animals were put into the testing box, where they could freely explore the environment for 10 min, and then 2 µL of concentrated 2-MT was pipetted onto a filter paper and placed on the lid inside the cage. The odor exposure was maintained for 10 min.

### 4.3. Perfusion and Tissue Processing

Mice were deeply anesthetized with ketamine–xylazine (intraperitoneal injection; ketamine, 83 mg/kg; xylazine, 3.3 mg/kg, Medicus Partner Kft., Biatorbágy, Hungary) and transcardially perfused with saline, followed by 70 mL of ice-cold fixative (4% formaldehyde in 0.1 M phosphate buffer, pH 7.2, Molar Chamicals Kft., Halásztelek, Hungary). The brains were then removed and post-fixed in the same fixative, supplemented with 10% sucrose (Molar Chamicals Kft., Halásztelek, Hungary), for 3 h. Subsequently, tissues were cryoprotected overnight in 10% sucrose in potassium phosphate-buffered saline (KPBS). Coronal brain sections (25 μm) were cut using a freezing microtome and divided into four series. The sections were stored at −20 °C in an antifreeze solution (30% ethylene glycol and 20% glycerol in 0.1 M PBS, Molar Chamicals Kft., Halásztelek, Hungary). The sections were mounted onto slides and coverslipped using DAPI Fluoromount-G (SouthernBiotech, Birmingham, AL, USA, Cat. No. 0100-20) as a nuclear counterstain for further examination and analysis. To visualize activated neurons, c-Fos immunohistochemistry was used. After washing in KPBS, the sections were incubated in 2% normal donkey serum (Jackson ImmunoResearch Europe Ltd., St. Thomas Place, UK) for 1 h at room temperature, then placed in rabbit anti-cFos IgG (sc-52, Santa Cruz Biotechnology, Santa Cruz, CA, USA (dilution: 1:10,000) and incubated overnight at 4 °C. The immunoreaction was visualized using donkey anti-rabbit IgG (ThermoFisher, Waltham, CA, USA, 1:1000)/Alexa Fluor 488 conjugate at room temperature for 2 h in dark. Sections were mounted onto slides and coverslipped with DAPI Fluoromount-G (SB. Birmingham, AL, USA, Cat. No. 0100-20) as a nuclear counterstain.

### 4.4. Imaging, Quantification and Data Analysis

Digital images of the brain sections were captured at 20× magnification using the 3D HISTECH Pannoramic MIDI II slide scanner. Regions of interest (ROIs) were outlined based on the Allen Brain Atlas and analyzed with NIS Elements Imaging Software (version 5.21.01).

### 4.5. Statistical Analysis

All quantitative data are presented as the mean ± SEM (standard error of the mean). Data analysis was conducted using GraphPad Prism software (version 10; San Diego, CA, USA). Unpaired *t*-tests were performed to assess differences originating from sexual dimorphism and stressor type (psychological or physiological), with a *p*-value < 0.05 considered statistically significant.

## Figures and Tables

**Figure 1 ijms-25-12004-f001:**
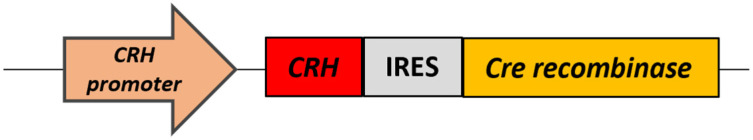
Concept of Crh-IRES-Cre mice. The expression of Cre recombinase is under the control of the CRH promoter.

**Figure 2 ijms-25-12004-f002:**
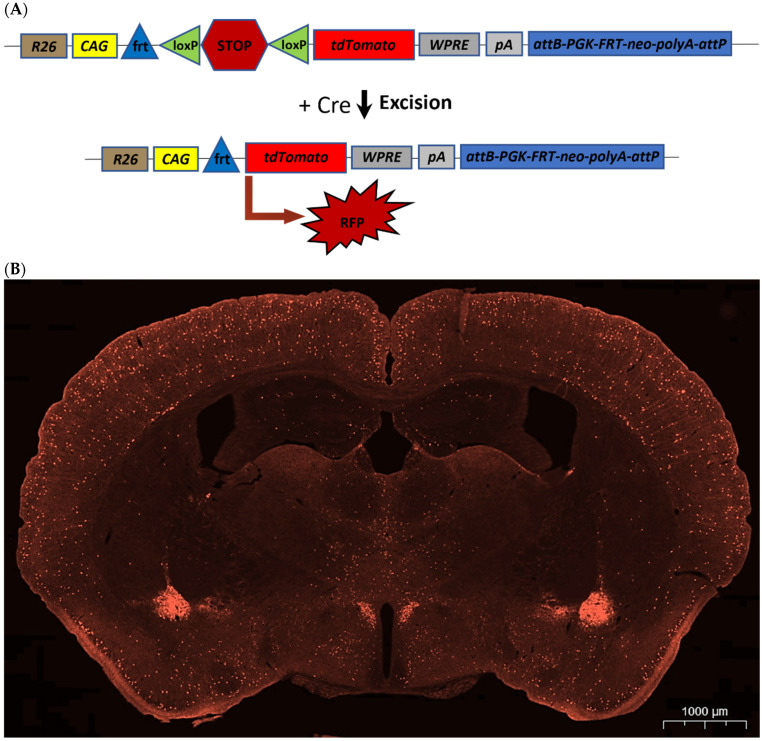
Crh-IRES-Cre;Ai9 mice. (**A**) In the presence of Cre recombinase, STOP codon excision occurs, which results in the expression of tdTomato red fluorescent protein (RFP) in Cre recombinase-containing cells. (**B**) Image of a Crh-IRES-Cre;Ai9 mouse brain section (25 μm thick) at the level of the mid-hypothalamus. Cre-mediated recombination results in red fluorophore expression in the CRH neurons. Note the intense signal in the hypothalamic paraventricular nucleus and central amygdala and scattered cells distributed along the cortical areas.

**Figure 3 ijms-25-12004-f003:**
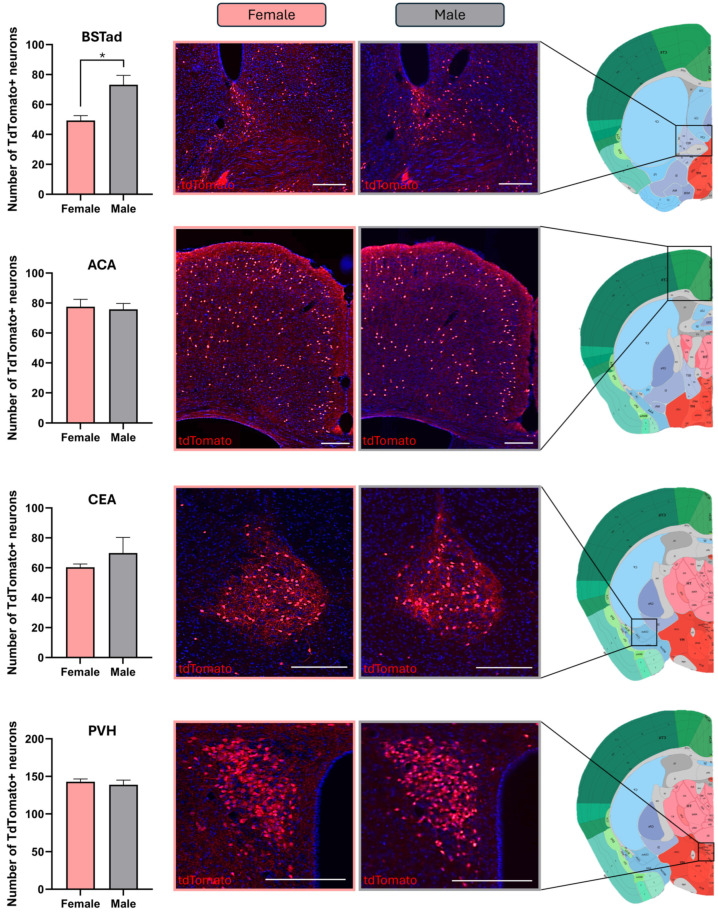
Presence of tdTomato immunoreactive cells in the brains of female and male Crh-IRES-Cre;Ai9 mice across four distinct regions. BSTad (bed nuclei of the stria terminalis, anterior division, dorsal part), ACA (anterior cingulate area), CEA (central amygdalar nucleus), PVH (paraventricular hypothalamic nucleus). The corresponding microscopic images for each region display tdTomato+ neurons alongside brain atlas images for spatial reference. The scale bar in each microscopic image represents 200 µm. The drawings on the right are sourced from the online Allen Brain Atlas. * *p* < 0.05 for female vs. male mice.

**Figure 4 ijms-25-12004-f004:**
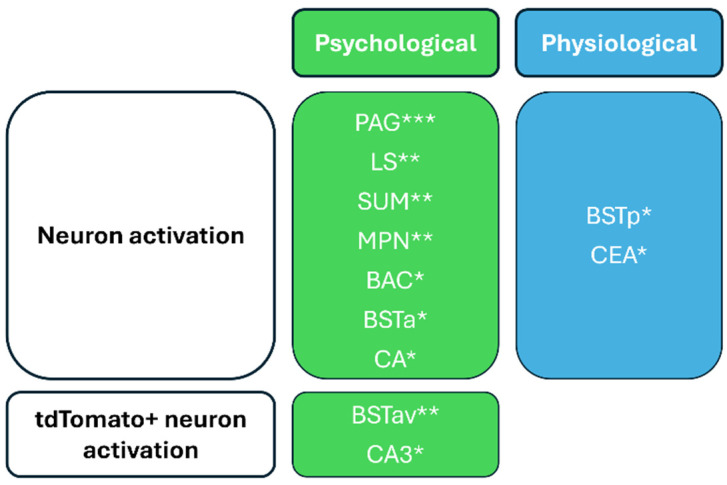
Comparison of elevation of neuronal activation and tdTomato+ neuron activation across various brain regions in response to psychological and physiological stressors. The left column lists brain regions that exhibited a significantly higher elevation of neuronal activation due to psychological stress than due to physiological stress, including the PAG (periaqueductal gray), LS (lateral septal nucleus), SUM (supramammillary nucleus), MPN (medial preoptic nucleus), BAC (bed nucleus of the anterior commissure), BSTa (bed nuclei of the stria terminalis, anterior division), and CA (Ammon’s horn). Additionally, as a result of psychological stress, significantly greater elevation in tdTomato+ neuron activation was detected in the BSTav (bed nuclei of the stria terminalis, anterior division, ventral part) and CA3 (field CA3). The right column highlights brain regions that showed significantly higher increases in neuronal activation in response to physiological stress than psychological stress, specifically the BSTp (bed nuclei of the stria terminalis, posterior division) and CEA (central amygdalar nucleus). * *p* < 0.05; ** *p* < 0.01; *** *p* < 0.001 for elevation of the activation level due to psychological stressors vs. physiological stressors. (Elevation of the activation level was defined as the fold change in treated mice compared to control animals). This figure illustrates the distinct neural responses to psychological versus physiological challenges.

**Figure 5 ijms-25-12004-f005:**
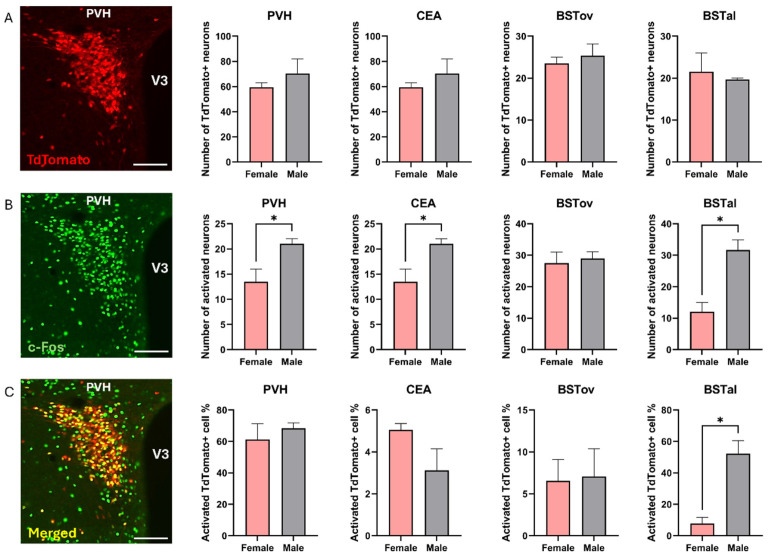
Comparison of recruitment of CRH neurons in female and male mice exposed to predator odor. Female and male Crh-IRES-Cre;Ai9 mice were exposed to fox odor for 10 min. Sex differences in the numbers of tdTomato+ cells (**A**), predator odor-activated c-Fos positive neurons (**B**), and predator-odor activated tdTomato+ (putative CRH) neurons (**C**) in the PVH (paraventricular hypothalamic nucleus), CEA (central amygdalar nucleus), BSTov (bed nuclei of the stria terminalis, anterior division, oval nucleus), and BSTal (bed nuclei of the stria terminalis, anterior division, anterolateral area). In the first column, representative images of tdTomato and c-Fos staining along with a merged image of the PVH can be seen. The scale bar in each microscopic image represents 100 µm. V3 (third ventricle). * *p* < 0.05 for female vs. male mice.

## Data Availability

The original contributions presented in the study are included in the article; further inquiries can be directed to the corresponding author.

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
