# Peer review of "Sex Differences in the Neuroendocrine Stress Response: A View from a CRH-Reporting Mouse Line"

_ijms, 2024, doi:10.3390/ijms252212004_

Round 1
Reviewer 1 Report
Comments and Suggestions for Authors
Thanks for the nice work.
My questions:
1. Authors might want to revise the sentence at line 396.
2. Do authors agree that though men usually have a higher cortisol response to stress than women, this can change based on the type of stress?
3. Why did authors consider only CRH-expressing neurons? What about oxytocin releasing neurons?
Author Response
Answer to REVIEWER #1:
We thank the reviewer for evaluating our work and for the comments. Here are the answers to the issues raised. We addressed all the points you raised and revised the manuscript as suggested.
1. Authors might want to revise the sentence at line 396.
Both reviewers commented the final conclusion of the manuscript. Accordingly, we have provided a more detailed, more explanatory conclusion paragraph.
“In summary, our present study reveals that tdTomato expression in CRH-Ires-CRE;Ai9 mice does not precisely mirror the sex differences in CRH mRNA, which develop in the brain of adult male and female mice under the influence of sex steroids during ontogenesis and after puberty. The presence of CAG promoter in CRH-Ires-CRE;Ai9 mice results in constitutive expression of the tdTomato marker gene, irrespective of the transcriptional status of the CRH gene. Nevertheless, we found more c-Fos immunoreactive neurons in the brain stress circuitry in predator odor exposed males compared to females. Furthermore, in the male BNST more CRH neurons express activation marker c-Fos than in females, indicating sex differences in organization of neuroendocrine, autonomic and behavioral responses to predators. “
2. Do authors agree that though men usually have a higher cortisol response to stress than women, this can change based on the type of stress?
We do not fully agree. Indeed, in an elegant study by Reschke-Hernandez et al (PMID: 27870432) using experimental psychological stress situation, such as Trier Social Stress Test and Iowa Singing Social Stress Test (public singing) men showed greater increase of cortisol compared to females. The study does not report however the differences in baseline cortisol levels, which are usually significantly higher in females. Thus the overall cortisol output might be higher in females compared to males. Another study categorized cortisol responses in men as no change, increase, and decrease and found correlation between cortisol stress response and facial asymmetry, competitiveness and copying strategy (https://doi.org/10.1080/10253890.2017.1378341). By contrast, studies on really “tough” men including those sentenced because of violent crimes display even dampened cortisol response (https://doi.org/10.1016/j.yhbeh.2022.105260). Low stress system reactivity in these individuals -in combination with high testosterone plasma levels- is associated with decreased affect and lack of concern for distress in others. From our c-Fos data (and colocalization of c-Fos in tdTomato (putative CRH neurons) seems that in response to predator odor, more CRH neurons are recruited in the male BNST, supporting the sex difference of stress reactivity.
3. Why did authors consider only CRH-expressing neurons? What about oxytocin releasing neurons?
We agree with the reviewer, that several additional factors and neuropeptides are involved in the sexual dimorphism associated with stress. In this respect, oxytocin plays a pivotal role. However, the aim of our present work was to focus on the role of CRH neurons, exploiting the CRH-IRES-Cre Ai9 reporter mice. Further research may clarify the interplay between this two (antagonistic) neuropeptides on the organization of HPA axis responses to stress.
Reviewer 2 Report
Comments and Suggestions for Authors
The aim of the study was to investigate whether stress related neurons and CRH cells are recruited in a stressor- on sex-specific manner within and/or outside the hypothalamus.The responses to psychological and physiological stressors were observed in specific regions of mice brain.
As an animal model for testing the distribution of CRH-expressing neurons in the adult mouse brain, Crh-IRES-Cre;Ai9 reporter mice were used.
It was found that that the number of tdTomato+ neurons was slightly higher in male mice within the bed nuclei of the stria terminalis, anterior division, dorsal part (BSTad) compared to female animals. Unfortunately,no sex-dependent differences in the numbers of CRH neurons were observed in the rest examined parts.
The Introduction, Material and Methods, Results and Discussion are well, logically written, with details.
I would suggest to write more detailed conclusions and avoiding to recall so many times the results of previous publication. Sometimes, it seems that Authors mixed the previous and present results. Please, carefully check the list of the tested areas.
Also, maybe the title should be changed-there was no clear sex differences in many tested regions.
Author Response
It was found that that the number of tdTomato+ neurons was slightly higher in male mice within the bed nuclei of the stria terminalis, anterior division, dorsal part (BSTad) compared to female animals. Unfortunately, no sex-dependent differences in the numbers of CRH neurons were observed in the rest examined parts.
Answer: Thank you for this remark. We think that one of the conclusion of our manuscript is that the sex differences, which are well described in the CRH stress-related circuitry, are not (could not be) revealed by using Crh-IRES-Cre;Ai9 reporter mice. We have included a more explanatory paragraph into the discussion:
“In our study, CRH neurons were identified using Cre-reporter transgenic mice (Crh-IRES-Cre;Ai9), in which a CAG-LoxStopLox-tdTomato construct is integrated into Gt(Rosa)26Sor locus (43). Since Cre recombinase and the CAG promoter are active throughout ontogenesis, it is likely that neurons active during embryonic development or in early life stages, but not in adulthood, also express the marker. Thus, although the rising testosterone levels during the critical developmental period may have reduced the number of CRH neurons in the BNST of male model animals, this reduction remain undetectable in our study. Our results implicate that CRE-based reporter mouse lines (ie. Crh-IRES-Cre;Ai9) identify all neurons, which have ever expressed the gene of interest (CRH), are useful for anatomical localization studies, however are not suitable for assessing the current transcriptional activity of that gene. “
I would suggest to write more detailed conclusions and avoiding to recall so many times the results of previous publication. Sometimes, it seems that Authors mixed the previous and present results. Please, carefully check the list of the tested areas.
Answer: In our previous paper, we tested 95 brain areas in male Crh-IRES-Cre;Ai9 mice, among which 23 were significantly stress related. These areas were then compared in this manuscript using male vs. female animals. Some representative examples are shown on Figure 3. We thought that the inclusion of paragraph 2.3 (in which we really recall the results of our previous results on male mice, but from another aspect (ie. comparison of psychological vs. physiological stress reactions) would support our intention to compare the effect of predator odor stress in male and female animals.
We have changed the conclusion paragraph with more detailed description:
“In summary, our present study reveals that tdTomato expression in CRH-Ires-CRE;Ai9 mice does not precisely mirror the sex differences in CRH mRNA, which develop in the brain of adult male and female mice under the influence of sex steroids during ontogenesis and after puberty. The presence of CAG promoter in CRH-Ires-CRE;Ai9 mice results in constitutive expression of the tdTomato marker gene, irrespective of the transcriptional status of the CRH gene. Nevertheless, we found more c-Fos immunoreactive neurons in the brain stress circuitry in predator odor exposed males compared to females. Furthermore, in the male BNST more CRH neurons express activation marker c-Fos than in females, indicating sex differences in organization of neuroendocrine, autonomical and behavioral responses to predators. “
Also, maybe the title should be changed-there was no clear sex differences in many tested regions.
Answer: Because we did find significant increase in the number of c-Fos positive cells and neurons which express both c-Fos and tdTomato+c-Fos markers in selected brain areas in males compared to females, we do not think to change the title of the manuscript, if the reviewer agrees.